# Preparation, Characterization and Pharmacokinetics of Tolfenamic Acid-Loaded Solid Lipid Nanoparticles

**DOI:** 10.3390/pharmaceutics14091929

**Published:** 2022-09-13

**Authors:** Wei Xu, Zhaoyou Deng, Yifei Xiang, Dujuan Zhu, Dandan Yi, Yihao Mo, Yu Liu, Lanqian Qin, Ling Huang, Bingjie Wan, Liqin Wu, Xin Feng, Jiakang He

**Affiliations:** 1College of Animal Science and Technology, Guangxi University, Nanning 530005, China; 2Department of Pharmaceutics and Drug Delivery, School of Pharmacy, The University of Mississippi, University, MS 38677, USA

**Keywords:** tolfenamic acid, solid lipid nanoparticles, suspension, sustained release, bioavailability

## Abstract

The clinical use of nonsteroidal anti-inflammatory drugs is limited by their poor water solubility, unstable absorption, and low bioavailability. Solid lipid nanoparticles (SLNs) exhibit high biocompatibility and the ability to improve the bioavailability of drugs with low water solubility. Therefore, in this study, a tolfenamic acid solid lipid nanoparticle (TA-SLN) suspension was prepared by a hot melt–emulsification ultrasonication method to improve the sustained release and bioavailability of TA. The encapsulation efficiency (EE), loading capacity (LC), particle size, polydispersity index (PDI), and zeta potential of the TA-SLN suspension were 82.50 ± 0.63%, 25.13 ± 0.28%, 492 ± 6.51 nm, 0.309 ± 0.02 and −21.7 ± 0.51 mV, respectively. The TA-SLN suspension was characterized by dynamic light scattering (DLS), fluorescence microscopy (FM), scanning electron microscopy (SEM), differential scanning calorimetry (DSC), and Fourier transform infrared (FT-IR) spectroscopy. The TA-SLN suspension showed improved sustained drug release in vitro compared with the commercially available TA injection. After intramuscular administration to pigs (4 mg/kg), the TA-SLN suspension displayed increases in the pharmacokinetic parameters T_max_, T_1/2_, and MRT_0–∞_ by 4.39-, 3.78-, and 3.78-fold, respectively, compared with TA injection, and showed a relative bioavailability of 185.33%. Thus, this prepared solid lipid nanosuspension is a promising new formulation.

## 1. Introduction

The welfare of animals has received increasing attention in recent years. The World Organization for Animal Health (OIE) develops and implements animal welfare standards, and new standards are constantly being added [1]. The reduction and prevention of pain are important factors in animal welfare; however, various conditions in farm animals, including pigs, can cause pain and inflammation [2]. Nonsteroidal anti-inflammatory drugs (NSAIDs) are the main drugs used to treat fever, pain, and inflammation in animals [3]. Most of the commonly used NSAIDs are not easily soluble in water and are classified as Class II drugs in the Biopharmaceutical Classification (BCS); tolfenamic acid (TA) belongs this class [4]. Poor water solubility affects drug absorption in the body during treatment, resulting in low bioavailability [5]. Moreover, NSAIDs cause side effects and gastrointestinal irritation after administration [6].

TA, which has the chemical name 2-[(3-chloro-2-methyl phenyl)-amino] benzoic acid (Figure 1), is a widely used NSAID [7] that can be administered to both humans and animals. Compared to other drugs of the same kind, NSAIDs exhibit a better effect and fewer side effects [8] and have a broad market at home and abroad. TA has been widely approved for antipyretic, analgesic, and anti-inflammatory treatment of cattle, pigs, dogs, cats, and other animals. It is also used for the treatment of mastitis and respiratory infections in cattle and as an adjuvant treatment of mastitis and diarrhoea syndrome [9]. The main mechanism of action of TA is the prevention of COX enzyme formation by prostaglandins through the inhibition of prostaglandin production throughout the body, which reduces inflammation, pain, and fever [10,11]. TA is rapidly absorbed and eliminated by the body with a short half-life (1.94–5.71 h) [12,13]. Currently, there are only two common dosage forms of TA available on the market, i.e., tablets and injections. However, these existing preparations cannot solve the problems of its unstable absorption and short half-life; notably, the half-life of the commercially available injection is 2.97 h (2.68–3.22 h) following intramuscular administration [13]. Therefore, to solve the problems with the existing drugs, suitable long-acting sustained release preparations must be developed. To date, there is no related literature report on the pharmacokinetics of TA in pigs. Thus, it is necessary to study the pharmacokinetics of TA in pigs to provide a basis for clinical trials.

Solid lipid nanoparticles (SLNs) are a new type of nanocarrier system composed of natural or synthetic solid lipids as the backbone material that have a high melting point. Drugs are encapsulated or embedded in the lipid core, and the system has a particle size of 50–1000 nm. This solid lipid particle drug delivery system has good biocompatibility and excellent physical and chemical properties. This system also has several advantages, including high physical stability, reduced drug leakage, good sustained release properties, and low toxicity; additionally, it can be easily mass produced [14]. SLNs are mainly composed of biocompatible solid lipids, surfactants, and drugs [15]. Solid lipids are generally those with a high-melting matrix, and trioleic acid, cholesterol, stearic acid, palm wax, and glyceryl monostearate are commonly used. Generally, polyethylene glycol, Tween-80, polyvinylpyrrolidone, and polyvinyl alcohol (PVA) are used as surfactants, which mainly provide stability and assist with encapsulation [16]. The drug is usually loaded in a lipid matrix, which is encapsulated by a nontoxic active agent. The choice of lipids and surfactants plays a key role in the formation of SLNs. Lipids and surfactants also affect the particle size, morphology, zeta potential, encapsulation efficiency (EE), drug loading, long-term stability, and in vitro drug release rate of SLNs [17]. Barbosa et al. used SLN technology to effectively prolong the anaesthesia effects of drugs, and Zhu et al. used SLNs to increase the bioavailability of drugs by sixfold. SLNs can be administered in various ways, such as oral, injection, eye, skin, and lung routes. The methods of preparing SLNs include high-pressure homogenization [18], microemulsion [19], solvent emulsification–evaporation [20], solvent emulsification–diffusion [21], hot melt–emulsification ultrasonication [22], thin film–ultrasonic dispersion [23] and solvent injection methods [24]. SLNs have the following characteristics. First, the solid lipids can control and maintain the drug release properties by decreasing the fluidity of the drug within the lipid core, thereby enhancing sustained release [25]. SLNs allow for targeted for drug delivery because they can hide the properties of the carried drug and then release it under controlled conditions to the desired targeted area. SLNs also protect the drug from the chemical and biodegradation associated with the route of administration [26]. Finally, SLNs can improve the drug release characteristics and speed up the drug dissolution process, thereby enhancing drug absorption and rerouting drugs through the lymphatic circulation to improve drug bioavailability [27]. Oral administration of TA may cause certain side effects in the gastrointestinal tract. Therefore, in this study, the characteristics of SLNs were exploited to prepare a TA suspension to reduce its side effects [28].

Herein, stearic acid was used as the solid lipid matrix and PVA was used as the surfactant to prepare the TA-SLN suspension by the hot melt–emulsification ultrasonication method. Ultrasound is one of the most effective methods to prepare SLNs. The ultrasound waves generated by ultrasound devices provide unique reaction conditions and are widely used to prepare various nanomaterials [29]. It is generally believed that the size of the drug molecule should be smaller than the length of the ultrasound to prevent a direct interaction [30]. Ultrasound can provide a large amount of energy locally through cavitation [31], so the lipid carrier can be created on the nanometre scale. This study provides a detailed evaluation of the physical properties, in vitro release, and pharmacokinetics of the most effective TA-SLN suspension.

## 2. Materials and Methods

### 2.1. Materials

The TA standard (purity 99.70%) was obtained from Dr. Ehrenstorfer GmbH, Co., Ltd. (Shanghai, China). TA (purity 99.84%) was obtained from Shanghai Bank Pharmaceutical Technology Co., Ltd. TA injection (50 mL, 2 g) was purchased from Hebei Weiyuan Pharmaceutical Co., Ltd. Glyceryl monostearate was purchased from Sinopharm Chemical Reagent Co., Ltd. Compritol^®^ 888ATO was purchased from Garfax (France). Stearic acid and carnauba wax were purchased from Shanghai Yuanye Biological Technology Co., Ltd. Poloxamer 188 and polyvinylpyrrolidone K_30_ (PVP_K30_)-were obtained from Beijing Soleibao Technology Co., Ltd. PVA (molecular weight: 16,000) was obtained from Macklin (Shanghai China). Phosphoric acid (H_3_PO_4_), potassium chloride (KCl), and potassium dihydrogen phosphate (KH_2_PO_4_) were purchased from Tianjin Damao Chemical Reagent Factory. Sodium chloride (NaCl) was purchased from Chengdu Jinshan Chemical Reagent Co., Ltd. Disodium hydrogen phosphate dodecahydrate (Na_2_HPO_4_·12H_2_O) was purchased from Guangdong Guanghua Technology Co., Ltd. Ultrapure water for all experiments was obtained with a Microporous Purification^®^ Purified Water System (CD-UPT-Ⅱ-20, Chengdu Yuechun Technology Co., Ltd.).

### 2.2. Animals

Six healthy 30-day-old binary pigs (Landrace × Dabai), half male and half female weighing 15 ± 2 kg were purchased from Guangxi Dazhong Farm and reared at the test site of Guangxi University after one week of adaptation to the environment. During the adaptation period, the temperature of the breeding environment was 18–25 °C, the relative humidity was 50–65%, and drinking water was freely accessible. Feed without any medicine was added during the breeding period. All animal experiments were approved by the Animal Ethics Committee of Guangxi University (approval number GXU 2020-035).

### 2.3. Preparation of the TA-SLN Suspension

The TA-SLN suspension preparation process is shown in Figure 2. The suspension was prepared by the hot melt–emulsification ultrasonication method. Briefly, 2 g of TA and 4 g of stearic acid were placed in a beaker and heated with magnetic stirring at 150 rpm. When the solid lipid and drug had completely melted, 30 mL of 2% PVA was added at the same temperature, and the mixture was stirred with a glass rod to form an O/W colostrum. Subsequently, an ultrasonic cell (SCIENTZ-ⅡD, 900 W, 20 KHz, Ningbo Xinzhi Biotechnology Co., Ltd.) Disrupter was used for 8 min to form a nanoemulsion. The nanoemulsion was quickly poured into sterilized water at 4 °C, and the volume was adjusted to 100 mL to obtain a SLN suspension, which was disinfected with cobalt 60 gamma rays.

### 2.4. Characterization of the TA-SLN Suspension

#### 2.4.1. Determination of Particle Size, Polydispersity Index (PDI), and Zeta Potential

A laser particle size analyser (ZSE; Malvern Panalytical, Co., Ltd., Worcestershire, UK) was used to measure the particle size, PDI, and zeta potential of the sample at 25 °C. The suspensions were diluted 100-fold with distilled water for measurement. All samples were measured in triplicate.

#### 2.4.2. Loading Capacity (LC) and EE

The TA-SLN suspension was centrifuged at 12,000 rpm (Hitachi Centrifuge CF16RN; Hitachi Koki Co., Ltd. Tokyo, Japan) at 4 °C for 60 min, the supernatant was discarded, and the precipitate was washed 3 times with distilled water. Subsequently, the supernatant was removed, and the SLN precipitate was resuspended in distilled water and dried with a freeze dryer for 48 h (FD-1, Freeze Dry System; Beijing Bo Yikang Experimental Instrument Co., Ltd.) to obtain a freeze-dried powder. A total of 10 mg of the lyophilized powder was weighed into a 15 mL EP tube, 10 mL of a mixed solution of acetonitrile and water (1:1) was added, and the EP tube was placed into a water bath to destroy the nanoparticles and release the encapsulated drug. Then, a mixed solution of acetonitrile and water was used to determine the volume, and the volume was adjusted to 10 mL before centrifugation at 8000 rpm for 10 min. The supernatant was diluted and analysed with a Waters e2695 series HPLC equipped with a Waters 2998 PDA detector (Waters, Milford, MA, USA). The formulas for calculating the EE and drug loading are as follows:EE (%) = (Weight of TA in the SLNs)/(Weight of TA added) × 100%
LC (%) = (Weight of TA in the SLNs)/(Weight of SLNs) × 100%

#### 2.4.3. Morphology Observation

The morphology of the TA-SLN suspension was observed by optical microscopy and scanning electron microscopy (SEM). Two microlitres of diluted sample was placed on a glass slide and dried naturally at room temperature before being observed and photographed under an optical microscope (TS2-FL, Nikon, Japan). For the SEM observations (FEI Quattro S, Thermoelectric, USA), the samples were naturally dried at room temperature for 5 min, dried in an oven, and sprayed with gold under a 20 mA current. After gold spraying, the morphology of the samples was observed by SEM.

#### 2.4.4. Differential Scanning Calorimetry (DSC)

Taking 3–10 mg of the sample and using alumina as the reference, a differential scanning calorimeter (DSC404F3, Netesch, Germany) was used to measure the parameters in the heating range of 30–300 °C at a heating rate of 10 °C/min in an inert gas environment at 20 mL/min to obtain a thermal curve.

#### 2.4.5. Fourier Transform Infrared (FT-IR) Spectroscopy

FT-IR spectroscopy was recorded using an FT-IR (Frontier, PE, USA) emission spectrometer. TA raw materials, excipients, and the TA-SLN samples were examined by IR spectroscopy using the potassium bromide tableting method. The determination resolution was 4 cm^−1^, and the wavelength range was 4000–400 cm^−1^.

#### 2.4.6. Settlement Rate, Redispersibility, and pH

The nanosuspension was added to a 50 mL stoppered graduated cylinder, diluted to the mark and shaken vigorously to obtain a uniform suspension. The height (H_0_) was measured at this time, and after standing for 3 h, the final height of the sediment was recorded as H. The sedimentation volume ratio was recorded as F by the formula F = H/H_0_.

The nanosuspension was placed in a 100 mL stoppered measuring cylinder and was left to stand for 24 h after the cylinder was tightly plugged. After inverting the measuring cylinder, the cylinder was forcefully turned upright (one positive and one inverse turn counted as one turn), and the number of turnings necessary for the sediment at the bottom of the measuring cylinder to disappear was recorded. This method better reflects the redispersibility than other methods.

The pH of the TA nanosuspension was tested with a pH meter (PHS-3c, Shanghai Precision Scientific Instrument Co., Ltd.) according to the instructions.

#### 2.4.7. Stability Studies

The stability of the TA-SLN suspension was evaluated by influencing factor experiments, including high temperature, high humidity, and strong light. The suspension was placed in a 40 °C environment (high-temperature test), at 25 °C with 90% ± 5% humidity (high humidity test), or exposed to 4500 ± 500 lx (high light test) for 10 d.

### 2.5. In Vitro Release

To determine the release of the TA-SLN suspension in pH 7.4 buffer, the 2015 edition of the “Veterinary Pharmacopoeia of the People’s Republic of China” appendix “Dissolution and Release Determination Method” paddle method was referenced. A total of 500 mL of degassed pH 7.4 buffer solution was added to the dissolution vessel (RC-3, Tianjin Tianguang New Optical Instrument Co., Ltd.). A total of 1 mL of the TA-SLN suspension (containing 20 mg TA) was removed, placed in a dialysis bag (molecular weight: 20,000), and put into a dissolution vessel. Additionally, 1 mL of TA injection was placed in a dialysis bag that was put in a dissolution vessel as a control. The dissolution temperature was set to 37 ± 0.5 °C, and the rotational speed was 50 r/min. Sampling was performed at fixed time points, 1 mL was removed each time, and then the same temperature and volume of pH 7.4 buffer was added into the dissolution vessel after sampling. Each sample was filtered through a 0.22 μm filter. The concentration of the sample was determined by HPLC, and a plot of the in vitro release profile was constructed.

### 2.6. Pharmacokinetics

After one week in the experimental setting, 6 pigs were randomly divided into two groups of three pigs, labelled as sizes 1–6, and weighed. The animals were fasted for 12 h before administration with free access to drinking water. The TA-SLN suspension was sterilized by cobalt 60 gamma rays before administration. Both the TA-SLN suspension and commercial TA injection were administered intramuscularly at 4 mg/kg. A two-cycle crossover experiment was performed for 6 pigs, with each crossover cycle separated by one week. Before administration (0 h) and at different time points after administration (0.16, 0.25, 0.5, 0.75, 1, 2, 4, 6, 8, 12, 24, 36, 48, 72, 96 and 120 h), 5 mL of blood was collected from each pig’s anterior vena cava and centrifuged at 3000 r/min for 10 min to separate plasma. Plasma samples were extracted by acetonitrile precipitation, and the extraction procedure was repeated once. Then, the combined supernatant was dried with nitrogen, and the residue was dissolved in mobile phase. The drug concentration in the plasma was determined by HPLC after pretreatment.

According to the blood drug concentration–time data from each pig measured in the experiment, an average drug–time curve was drawn for each group. WinNonlin version 5.2.1 was used to calculate the main pharmacokinetic parameters of the two drugs (T_1/2_, T_max_, C_max_, AUC_0–∞_, Vd, MRT_0–∞_ and CL) and fully reflect the characteristics of the two drugs, including absorption, distribution, and elimination, in pigs.

### 2.7. Determination of TA in Plasma Samples

The concentration of TA in each sample was measured using HPLC. The chromatographic conditions were as follows: the column was an Inertsil ODS-3 C18 (5 µm, 4.6 × 250 mm), the mobile phase was 0.3% phosphoric acid in water as the aqueous phase and acetonitrile as the organic phase, isocratic elution (70:30) was performed, and the detection wavelength was 289 nm. Blank plasma was added to the standard solution to prepare a standard working curve of plasma drug concentrations in the range of 0.05–10 µg/mL (R^2^ = 0.9995). The limit of detection in plasma was 0.05 µg/mL, and the limit of quantification was 0.1 µg/mL. The average recoveries of TA in plasma samples were 84.86–100.46%. The interassay coefficient of variation was less than 4.31%.

### 2.8. Statistical Analysis

Significant differences in the main pharmacokinetic parameters of the two preparations were analysed using SPSS Statistics 21 software (version 21, IBM, New York, NY, USA). Data are presented as the mean ± SD.

## 3. Results

### 3.1. Optimization of the TA-SLN Suspension

TA was found to be miscible with four different lipids at 100 °C. Its maximum solubility in Compritol^®^ 888ATO, glyceryl monostearate, stearic acid, and carnauba wax was 16.66%, 33.33%, 25.00%, and 33.33%, respectively. Since glyceryl monostearate reacts with acidic compounds, resulting in precipitation of the drug, this lipid was not considered. Additionally, considering cost, stearic acid was finally selected as the solid lipid, and the optimal drug-to-lipid ratio was 1:4. An L_9_ (3^4^) orthogonal test table was used to screen surfactants with drug loading as the investigation index. The orthogonal design and results are shown in Table 1 and Table 2. The order of influence of each factor on drug loading was A > C > B, among which the type of surfactant had the greatest influence, and the optimal combination was A_2_B_1_C_3_. Thus, 30 mL of 2% PVA was used as the surfactant.

### 3.2. Properties of the TA-SLN Suspension

Optical microscopy observations showed that the prepared TA-SLN suspension exhibited a uniform particle distribution and no aggregation (Figure 3A). SEM observations showed that the prepared TA-SLN suspension was spherical and uniformly dispersed (Figure 3B). The optimal SLN suspension particle size, zeta potential and PDI were 492.33 ± 6.51 nm, −21.7 ± 0.51 mV and 0.309 ± 0.02, respectively. The optimal drug loading and EE of the SLN suspension were 25.13 ± 0.28% and 82.50 ± 0.63%, respectively. The concentration, sedimentation rate and pH value of the TA-SLN nanosuspension were 2%, 1 and 6.2, respectively. Thus, the TA-SLN suspension must only be turned to allow the bottom sediment to disappear for its even dispersion.

### 3.3. Thermal Characterization

The DSC results are shown in Figure 4. The DSC curve for TA shows an endothermic peak at 210 °C, indicating that the melting point of pure TA is 210 °C, and stearic acid displayed an endothermic peak at 69 °C, implying that the melting point of stearic acid is 69 °C. PVA has an endothermic peak at 290 °C, suggesting that the melting point of PVA is 290 °C. The endothermic peak of TA-SLN appeared at 69 °C, and no endothermic peak corresponding to the pure drug was observed in the TA-SLN sample, which indicated that a molecular form of the drug was dispersed in the lipid.

### 3.4. FT-IR Spectroscopy

The FT-IR results are shown in Figure 5. Solid lipid stearic acid has C-H stretching vibration absorption peaks at 2850 and 2980 cm^−1^, the peak at 1710 cm^−1^ is from a C=O stretching vibration, and that at 1464 cm^−1^ is a C-H bending vibration absorption peak. C-O stretching vibration absorption peaks appear at 1099 cm^−1^ [32]. The surfactant PVA has a C=O stretching vibration absorption peak at 1732 cm^−1^, and that at 2940 cm^−1^ is an asymmetric C-H stretching vibration absorption peak [33]. The characteristics of the TA active pharmaceutical ingredient (API) are as follows: the peak at 1267 cm^−1^ is -CH deformation, 1500 cm^−1^ is an -NH bending vibration, 1661 cm^−1^ is a C=O stretching vibration absorption peak, and 3340 cm^−1^ is the -NH stretching vibration peak [34,35,36]; these peaks were observed in the TA-SLN sample. From the characteristic peaks of the above lipids, surfactants, and API, these components exist in the SLN without undergoing any changes in their natural chemical structures.

### 3.5. Stability

The influencing factor test results are shown in Table 3. The TA-SLN suspension did not change significantly in appearance, content labelling amount, or particle size after 10 d in the high temperature, high humidity, or strong light irradiation tests.

### 3.6. In Vitro Release

The results from the in vitro release tests of TA injection and TA-SLN suspension in pH 7.4 buffer are shown in Figure 6. The commercially available TA injection released 80.01% of the TA in 2 h. The drug displayed a fast release rate and was almost completely released within 12 h. The TA-SLN suspension only released 71.85% of the drug in 72 h, and the release rate was slow; the drug was basically completely released in 120 h. By comparing the in vitro release curves of the two preparations, the solid lipid nanosuspension can significantly prolong the release time of TA. Table 4 shows the fitted equations for the release kinetics, including the cumulative release (Mt) versus time (t) of TA from the TA-SLN suspension and commercially available TA injection samples (the release kinetics fitting results of the TA released from the SLN and commercial injection samples are listed in Table 4). The release kinetics fitting results indicated that the release of TA from the SLN suspension followed zero-order kinetics, which also indicated that the active ingredient in the suspension could be released from the system at a constant rate, while the commercially available TA injection followed first-order kinetics [37].

### 3.7. Pharmacokinetics

Using the blood drug concentration data, the drug–time curves of TA injection and TA solid lipid nanosuspension were drawn as shown in Figure 7. The drug–time curves show that the peak concentration of the TA-SLN suspension was lower than that of the commercial injection, but the peak time, half-life, and drug-time area under the curve were higher than those of the commercial injection. The major mean pharmacokinetic parameters of the two preparations are shown in Table 5. The AUC, T_1/2_, and MRT_0–∞_ of TA injection were 23.26 ± 4.92 h·µg/mL, 8.49 ± 1.36 h, and 5.88 ± 0.94 h, respectively, and these parameters of the TA-SLN suspension were 43.11 ± 1.16 h·µg/mL, 32.13 ± 0.75 h, and 22.26 ± 0.52 h, respectively (Table 5). Compared with the commercially available TA injection, the T_max_, T_1/2_, and MRT_0–∞_ of the TA-SLN suspension increased by 4.39-, 3.78-, and 3.78-fold, respectively. These results showed that the absorption of the TA-SLN suspension was better than that of the TA injection and had a certain sustained release effect, which improved the bioavailability of TA.

## 4. Discussion

When preparing SLNs, the choice of lipid and surfactant is crucial and affects the particle size, zeta potential, drug loading, EE, and PDI. When selecting lipids, solid lipids with better biocompatibility are generally chosen. After consulting the relevant literature, it was found that many NSAIDs use Compritol^®^ 888ATO and glyceryl monostearate as the solid lipid excipients when preparing nanoformulations [12,38,39]. Therefore, the above two lipids were used as excipients in this experiment. After drug fusion with the lipid glycerol monostearate, when a surfactant was added, the drug separated until it solidified. After reviewing the literature, it was found that the glycerol monostearate was mixed with solid lipids. Acidic compounds are reactive and not suitable for use with other acidic compounds, so these were not considered. When Compritol^®^ 888ATO was selected as the solid lipid, the particle size was (694.41 ± 7.25) nm, and the prepared nanosuspension solidified after cooling with a PDI > 0.5. It is generally believed that when the PDI is >0.5, the nanoparticle size distribution is wide and unstable [40]. Therefore, in this study, stearic acid was finally selected as the solid lipid. Stearic acid is a long-chain fatty acid with increased lipophilicity, which makes drug encapsulation easier. Stearic acid is the main component of body fat. It has a relatively complete degradation pathway in the body and is safe and nontoxic [41]. Surfactants are mainly divided into ionic and nonionic types, of which cationic surfactants are the most toxic and nonionic surfactants are the least toxic [42]. Therefore, in this study, the nonionic surfactant PVA was selected to mainly assist with stability and encapsulation. The EE and drug LC are mainly related to the choice of solid lipids and increase with increasing carbon chain length in the fatty acid. This is because the higher the hydrophobicity of long-chain fatty acids is, the better the adaptability of lipophilic drugs [43,44]. In this study, it was found that with increasing surfactant concentration, the size of the nanoparticles decreased. A high concentration of surfactant reduced the surface tension of the SLNs and promoted the distribution of particles under the action of ultrasonic waves. Although high concentrations of surfactant reduced the size of nanoparticles, surfactant toxicity increases at high concentrations [45]. Moreover, it was found that with increasing drug concentration, the nanoparticle size also increased, causing the whole system to reach a saturated state, which is consistent with literature reports [46]. The magnitude of the zeta potential affects the stability and cellular uptake of nanoparticles and the electrostatic interactions of the substances in the body [47,48]. Nanoparticles with a high surface potential are more efficiently phagocytosed by macrophages. Small surface potential differences and cell line both have a significant effect on nanoparticle uptake. In vivo biodistribution demonstrates that rationally designed nanoparticles can selectively deliver drug-loaded nanomaterials to the site of action, thereby maximizing therapeutic efficacy and minimizing side effects [49,50]. SLNs are usually produced in the form of a suspension or dry powder; for ease of handling and cost, suspensions are preferred [51].

To avoid using organic solvents that leave solvent residues, the hot melt–emulsification ultrasonication method was used in this experiment to prepare the TA-SLN suspension. Only natural lipid materials, such as drugs, lipids, and surfactants, were used in this method. Particular attention should be placed on selecting the ultrasonic power and time. According to the Ostwald ripening effect, when the ultrasonic power and time continuously increase, the particle size increases accordingly [52]. Therefore, it is necessary to select the appropriate ultrasonic power and time to prepare SLNs. In general, using this method has several advantages, including simple operation, easy preparation, and no solvent residue.

Phase characterization can provide a better understanding of the physical properties of drugs [53]. The commonly used characterization methods are powder X-ray diffraction (PXRD), DSC, and FT-IR [54,55,56]. In this study, DSC and FT-IR characterization techniques were used to characterize each excipient and the prepared TA-SLN suspension. The DSC results showed that TA was dispersed in the solid lipids in its molecular form. The FT-IR results confirmed that TA existed in the SLN without any change in its natural chemical structure. During drug cultivation, phase characterization techniques are usually necessary to analyse the preparation and ensure that it exists in the carrier in a certain form. A favourable basis for subsequent experiments can be ensured only if the existing form of the drug can be determined.

The stability of a preparation refers to the chemical, physical and biological stability of the drug under reasonable storage and transportation conditions. If chemical decomposition occurs in the preparation, not only will the efficacy of the drug be reduced, but some substances generated by chemical changes may even have serious toxic side effects. To improve the quality of the preparation, ensure efficacy and safety, and improve the economic benefit, research on the stability of the preparation is very important. Factors affecting formulation stability include not only environmental factors but also formulation factors, such as excipients, pH, and ionic strength. Herein, the test results of influencing factors showed that high temperature, high humidity, and strong light exposure produced no significant effects on the preparation.

The aim of the in vitro release test was to better understand the biopharmaceutical characteristics of the preparation and predict the release and absorption of the drug in vivo. Because this preparation is a nanopreparation, the dynamic membrane dialysis method and paddle method from the 2015 edition of the Chinese Veterinary Pharmacopoeia were used in this study to measure the in vitro release of the drug. A buffer solution with a temperature of 37 ± 0.5 °C and a pH of 7.4 (physiological conditions in vivo) was chosen. In this study, the external release time of the prepared nanosuspension was longer than that of the commercial formulation. Commercial preparations involve ordinary injections without sustained release effects, while nanosuspensions exhibit sustained release, so the in vitro release rate from a nanosuspension is slower than that of ordinary preparations. However, this is mainly because fatty acids are used as solid lipids in the prepared nanosuspensions, and the fatty acids have a significant effect on the release. The longer the fatty acid carbon chain is, the slower the release rate because the lipophilicity of longer chain fatty acids is enhanced, thus improving drug retention [57].

After consulting the relevant literature, no reports on the pharmacokinetics of TA in pigs were found. Therefore, it is necessary to study the pharmacokinetics of TA in pigs to provide a basis for clinical trials. In this experiment, the pharmacokinetic behaviours of two preparations of TA, commercial TA injection and a homemade TA solid lipid nanosuspension, was evaluated in pigs. Through the fitted plots of the pharmacokinetic parameters, the C_max_ of the commercially available TA injection was higher than that of the nanosuspension, but the other pharmacokinetic parameters, including T_max_, AUC_0–∞_, MRT_0–∞_, and T_1/2,_ were lower than those of the nanosuspension with extremely significant differences. Among them, the half-life of the commercial TA injection (5.88 ± 0.94 h) was lower than that of the TA-SLN suspension (22.26 ± 0.52 h). The T_max_ of the commercially available injection was shorter, which may be because the commercial injection enters the interstitial body fluid after intramuscular injection and then diffuses into the capillaries [58], so the peak time is shorter. However, the T_max_ of the homemade nanosuspension was longer, mainly because the viscosity of the solid lipid nanosuspension is greater and the diffusion is slower at the injection site; thus, the surface area necessary for the release of TA from the SLN is reduced, and the absorption of TA is slower [59,60]. TA was dissolved in the SLN dispersion medium, then diffused in molecular form and was slowly released; therefore, the SLN can significantly prolong the action time of TA in vivo. The particle size and viscosity of the drug in the preparation affect its absorption and release. SLNs contain surfactants, which can increase the viscosity of the solution [61]. The drug release characteristics are controlled and maintained by reducing the fluidity of the drug within the lipid core, resulting in the slow release of the drug into the body. In conclusion, the solid lipid nanosuspension prepared in this study displayed a better absorption effect than that of the commercially available injection and exhibited a certain sustained release effect along with improved bioavailability.

## 5. Conclusions

Through single-factor and orthogonal design experiments, the optimal TA solid lipid nanosuspension was prepared by the hot melt–emulsification ultrasonication method. The TA-SLN suspension improved drug stability, sustained release, and bioavailability. The technology used in the current work showed better efficiency, a lower cost, and a simple operation, which allows for easier process control, quality management, and scale up, making it a promising technology in the pharmaceutical industry.

## Figures and Tables

**Figure 1 pharmaceutics-14-01929-f001:**
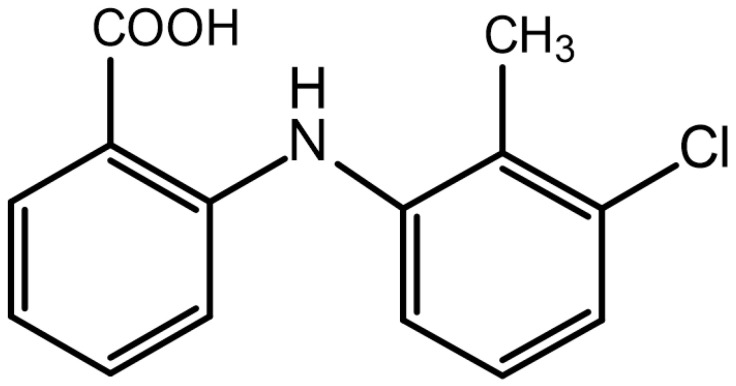
Structure of tolfenamic acid.

**Figure 2 pharmaceutics-14-01929-f002:**
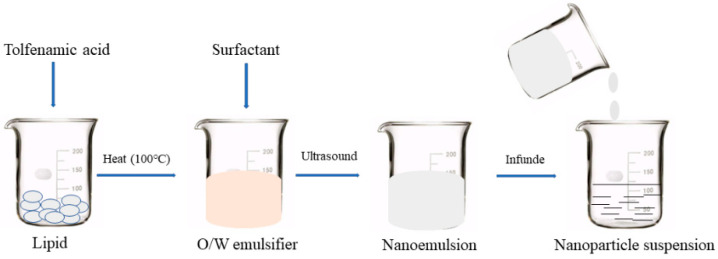
Process flow diagram.

**Figure 3 pharmaceutics-14-01929-f003:**
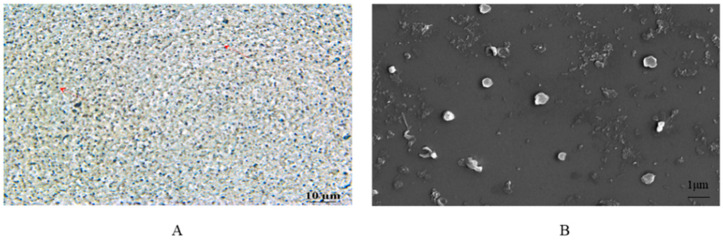
(**A**) Optical microscopy (magnification 40×) and (**B**) scanning electron microscopy (magnification 10,000×) photographs of the TA-SLN suspension.

**Figure 4 pharmaceutics-14-01929-f004:**
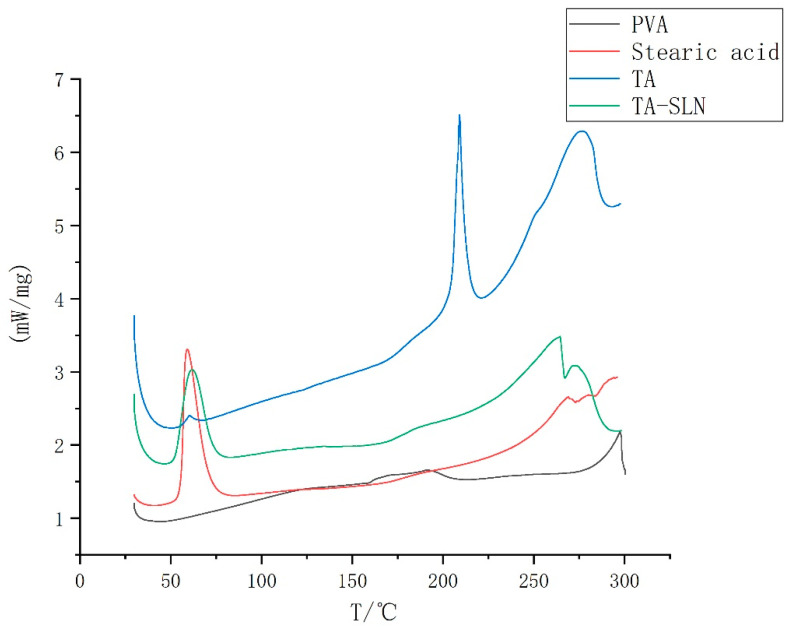
Differential scanning calorimetry (DSC) thermograms of PVA, stearic acid, TA and TA-SLN.

**Figure 5 pharmaceutics-14-01929-f005:**
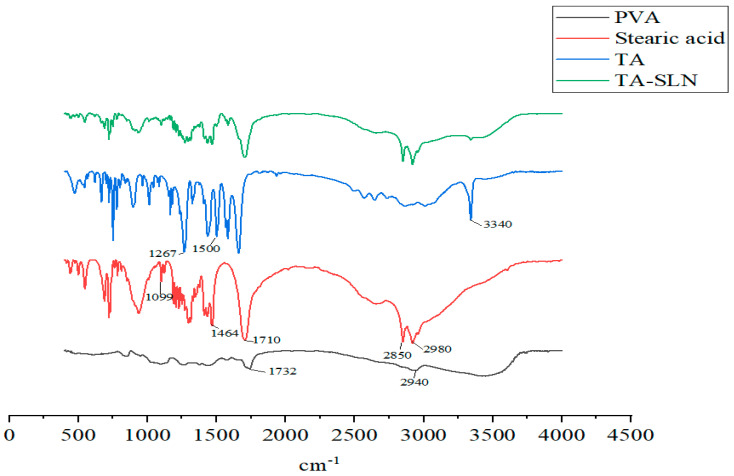
FT-IR analysis of PVA, stearic acid, TA and TA−SLN.

**Figure 6 pharmaceutics-14-01929-f006:**
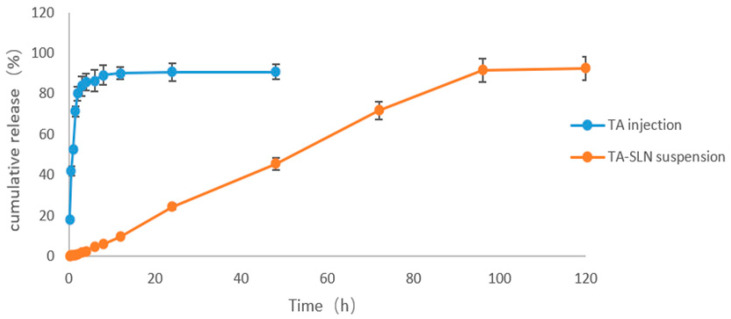
The time–release profiles of TA from TA injection and TA-SLN suspension in pH 7.4 PBS solution.

**Figure 7 pharmaceutics-14-01929-f007:**
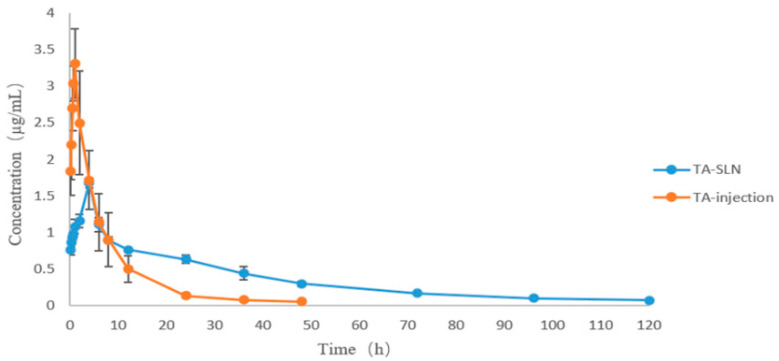
Concentration–time curves of the TA-SLN suspension and TA injection.

**Table 1 pharmaceutics-14-01929-t001:** Factor level table.

Variable	Level		
	1	2	3
Type	Poloxamer 188	PVA	PVP_K30_
Concentration	2%	3%	4%
Volume (mL)	20	25	30

**Table 2 pharmaceutics-14-01929-t002:** The design and results of the orthogonal experiment.

No	Type (A)	Concentration (B)	Volume (C)	LC (%)	Size (nm)	Zeta Potential (mV)	PDI
1	3	3	1	6.14 ± 0.12	417.63 ± 7.44	−24.17	0.27 ± 0.03
2	1	2	3	23.48 ± 0.58	458.83 ± 1.62	−22.77	0.35 ± 0.02
3	3	1	3	4.64 ± 0.09	577.51 ± 9.97	−25.43	0.40 ± 0.01
4	1	3	2	6.47 ± 0.14	508.53 ± 8.11	−22.60	0.27 ± 0.01
5	2	3	3	26.39 ± 0.37	668.30 ± 1.25	−11.17	0.34 ± 0.04
6	3	2	2	5.17 ± 0.14	1225.21 ± 5.29	−22.00	0.38 ± 0.04
7	2	2	1	16.59 ± 0.12	541.57 ± 4.72	−14.77	0.39 ± 0.02
8	2	1	2	26.61 ± 0.20	634.72 ± 4.81	−13.67	0.37 ± 0.01
9	1	1	1	20.81 ± 0.24	483.03 ± 9.17	−23.20	0.34 ± 0.01
K1	16.92	17.35	14.51				
K2	23.19	15.08	12.75				
K3	5.31	13.00	18.17				
R	17.88	4.35	5.42				
Optimum	A2	B1	C3				

**Table 3 pharmaceutics-14-01929-t003:** The influencing factors of the TA-SLN suspension (*n* = 3).

Influence Factors		Time (d)		
		0	5	10
High temperature	Appearance	Milky white	Milky white	Milky white
	Sedimentation rate	1	1	1
	Redispersibility	Good	Good	Good
	Content labelling amount (%)	100.19	99.36	99.22
	Relative substance	ND	ND	ND
	Size	460	482	479
High humidity	Appearance	Milky white	Milky white	Milky white
	Sedimentation rate	1	1	1
	Redispersibility	Good	Good	Good
	Content labelling amount (%)	100.19	99.22	99.83
	Relative substance	ND	ND	ND
	Size	460	466	471
High light	Appearance	Milky white	Milky white	Milky white
	Sedimentation rate	1	1	1
	Redispersibility	Good	Good	Good
	Content labelling amount (%)	100.19	99.04	99.35
	Relative substance	ND	ND	ND
	Size	460	485	491

ND, not detectable.

**Table 4 pharmaceutics-14-01929-t004:** Fitting equations of the cumulative release of TA versus time from the TA-SLN suspension and TA injection.

Fit Equation	TA Injection	TA-SLN Suspension
Zero-order equation	M_t_ = 66.15t + 0.78 (R^2^ = 0.1422)	M_t_ = 0.87t + 0.43 (R^2^ = 0.9792)
Formula One	M_t_ = 89.02 (1 − e^−1.043t^) (R^2^ = 0.9835)	-
Higuchi	M_t_ = 8.05X^1/2^ + 53.84 (R^2^ = 0.3648)	M_t_ = 9.66X^1/2^ − 14.01 (R^2^ = 0.9592)

**Table 5 pharmaceutics-14-01929-t005:** Pharmacokinetic parameters of TA in swine plasma after i.m. administration of TA injection and the TA-SLN suspension (*n* = 6).

Parameter	Unit	TA Injection	TA-SLN Suspension
T_max_	h	0.91 ± 0.21	4.00 ^b^
C_max_	µg/mL	3.37 ± 0.36	1.68 ± 0.07 ^b^
AUC_0–∞_	µg h/mL	23.26 ± 4.92	43.11 ± 1.16 ^b^
Vd	mL/kg	3220.21 ± 1323.89	4295.76 ± 508.01
CL	mL/h/kg	168.13 ± 27.11	85.60 ± 2.57 ^b^
MRT_0–∞_	h	8.49 ± 1.36	32.13 ± 0.75 ^b^
T_1/2_	h	5.88 ± 0.94	22.26 ± 0.52 ^b^

T_max_, time to reach maximum; C_max_, maximum concentration of drug; AUC_0–∞_, area under plasma concentration–time curve; Vd, volume of distribution; CL, clearance rate; MRT_0–∞_, mean residence time; T_1/2_, elimination half-life; F, bioavailability. ^b^ statistical significance compared with TA injection at *p* < 0.01.

## Data Availability

The data presented in this study are available on request from the corresponding author.

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
