# Peer review of "Preparation, Characterization and Pharmacokinetics of Tolfenamic Acid-Loaded Solid Lipid Nanoparticles"

_pharmaceutics, 2022, doi:10.3390/pharmaceutics14091929_

Round 1
Reviewer 1 Report
This research paper reported the synthesis and characterization of tolefenamic loaded solid lipid nanoparticles and their pharmacokinetics, which is an interesting and important area of research. The paper is generally well written and merits publication; however, the quality of the paper can be enhanced if the following points can be addressed.
1. In Line 60, it is mentioned that tolefenamic acid have a short life. Please cite the half life of the currently existing formulations.
2. In the discussion compare the authors need to compare the half life of the currently marketed formulations of tolefenamic acid with their formulations.
3. Did the authors perform any study to check the presence of surfactant after the synthesis of the nanoparticles? If no, please confirm
4. Mention the age or animals used for this study and the reason for employing pigs for this study.
5. I suggest the authors to mark the bands in FTIR with lines corresponding to each material for easier understanding.
6. Why only one pH was chosen for this study in drug release?
Reviewer 2 Report
[Pharmaceutics] Manuscript ID: pharmaceutics-1856540-peer-review-v1
Article
“ Preparation characterization and pharmacokinetics of tolfen-2 amic acid-loaded solid lipid nanoparticles”
Dear Authors,
While simple, the study has a robust initial thesis based on the inherent capability of solid lipid nanoparticles. The current research has valuable scientific points to offer. In addition, the study, while simple, has valuable scientific points. In addition, the measurement of bioavailability factors has been measured for the first time in pigs, so doing this part of the study seems challenging and difficult.
Comments:
1. Line 29, The way to use the suspension should be mentioned.
2. Line 42, Biopharmaceutical Classification of tolfenamic acid should be specified.
3. Line 67, “nanodrug system” should be “nanocarrier system”.
4. Line 76, “PVP and PVA” Should write the full name.
5. Line 106, “so the drug can reach the nanometre level”, Please change the word “drug” to ”lipid carrier”.
6. Line 118, Please mention the molecular weight of Polyvinyl alcohol (PVA).
7. Line 139, Please mention the rpm of stirring.
8. Line 161, Has the acetonitrile phase alone been used to measure the amount of drug? In this case, how is the amount of medicine that remains in the water phase calculated? To calculate EE and LC explain more.
9. Line 211, How much TA is in 1 mL of the TA- 211 SLN suspension? Please mention it.
10. In section “Pharmacokinetics” How did you ensure TA-SLN suspension was sterile for the injection?
11. What things were considered for the TA-SLN 223 suspension to be sterile during the process? These items should be mentioned.
12. Line 229, How the plasma sample was prepared? In more detail, refer to the stages of plasma preparation.
13. Line 292 and Line 308, Set the font of the words.
14. Line 332, Please mention the time of AUC.
15. Line 356, It is better to report the size of SLN made by Compritol® 888ATO, too.
16. Line 374, “, The toxicity of surfactants increases at high concentrations”. This applies to ionic surfactants and some non-ionic surfactants. The toxicity of PVA is very low.
17. Line 448, Since both the manufactured TA_SLN and commercial forms, are administered by injection, how has the better absorption of the suspension been tested?
Regards,
Reviewer 3 Report
The manuscript seems interesting, but some clarification is required:
1) The results will strongly depend on the power of ultrasonic treatment per unit volume of the system. Give this value or at least detailed information about the mode of ultrasonic treatment.
2) The release kinetics require a mathematical description. This can help work (doi 10.3390/ma14205977 doi 10.3390/polym13152569).
3) For such a detailed study, the conclusions should be expanded.
